# Investigating the Mechanical and Durability Characteristics of Fly Ash Foam Concrete

**DOI:** 10.3390/ma15176077

**Published:** 2022-09-01

**Authors:** Sheng Li, Hongbo Li, Changyu Yan, Yongfa Ding, Xuanshuo Zhang, Jing Zhao

**Affiliations:** 1College of Civil and Hydraulic Engineering, Ningxia University, Yinchuan 750021, China; 2Engineering Research Center for Efficient Utilization of Water Resources in Modern Agriculture in Arid Regions, Yinchuan 750021, China; 3Ningxia Research Center of Technology on Water-Saving Irrigation and Water Resources Regulation, Yinchuan 750021, China

**Keywords:** FAFC, compressive strength, thermal conductivity, pore structure, frost resistance

## Abstract

Although fly ash foam concrete (FAFC) is lightweight, heat-retaining, and insulating, its application options are constrained by its weak construction and short lifespan. The effects of various dosage ratios of the foaming agent (i.e., hydrogen peroxide), silica fume, and polypropylene fiber on the dry density, compressive strength, thermal insulation performance, pore structure parameters, and durability of FAFC were analyzed in this study, which sought to address the issues of low strength and low durability of FAFC. According to the findings, there is a negative correlation between the amount of hydrogen peroxide (as the foaming agent) and compressive strength, and, as the silica fume and polypropylene fiber (PP fiber) content rise, the strength will initially rise and then fall. The distribution of pore sizes gradually shifts from being dominated by small pores to large pores as the amount of foaming agent increases, while the porosity and average pore size gradually decrease. When the hydrogen peroxide content is 5%, the pore shape factor is at its lowest. The pore size distribution was first dominated by a small pore size and thereafter by a large pore size when the silica fume and PP fiber concentration increased. Prior to increasing, the porosity, average pore size, and pore shape factor all decreased. Additionally, the impact of PP fiber on the freeze–thaw damage to FAFC was also investigated at the same time. The findings indicate that the freeze–thaw failure of FAFC is essentially frost heave failure of the pore wall. The use of PP fiber is crucial for enhancing FAFC’s ability to withstand frost. The best frost resistance is achieved at 0.4% PP fiber content. In conclusion, the ideal ratio for overall performance was found to be 5% hydrogen peroxide content, 4% silica fume content, and 0.1% polypropylene fiber content. The results obtained could be applied in different fields, such as construction and sustainable materials, among others.

## 1. Introduction

In recent years, with China’s rapid economic and social development, infrastructure has also developed rapidly, energy consumption has accelerated, and the energy crisis has been expanding, which has caused widespread concern. To reduce the consumption of resources and alleviate the energy crisis, we can start by decreasing the energy consumption of buildings. The main advantages of foam concrete (FC) are its light weight, its excellent sound and heat insulation properties, high fire resistance, low price, and easier pumping and application [1]. The use of FC can also reduce carbon dioxide emissions [2,3], so it is widely used in vibration damping, insulation, mine backfilling, and structural seismic protection and other building facilities. The total production of fly ash and silica fume in the Ningxia region in 2021 reached 1800 × 10^4^ t. Its considerable accumulation not only occupies land but also causes severe environmental pollution, so the question of how to obtain value from waste and realize resource utilization is a problem worth studying. Research showed that the active substances SiO_2_, Al_2_O_3_, etc., in fly ash and silica fume can be used as a mineral admixture to promote the cement hydration reaction [4,5,6], which can save resources when mixed into cement slurry—for example, the use of fly ash can reduce the required cement dosage by 50% [7]. Adding fly ash and silica fume into foamed concrete represents the resource utilization of fly ash and silica fume, realizing the concept of obtaining value from waste, and realizing the requirements of environmental protection. The hydration heat of cement is high, which will adversely affect the environment. The addition of fly ash and silica fume can replace part of the cement. The reduction of the cement dosage can reduce the generation cost of foamed concrete, which is beneficial for environmental protection, and fly ash and silica fume can improve the performance of foamed concrete. Foamed concrete not only has the characteristics of light weight, good fire resistance, good sound insulation, good thermal insulation, and good seismic performance, but also has excellent performance, such as obtaining value from waste, environmental protection and energy savings, a simple production process, and low production cost. It is widely used in building insulation materials, garden decoration, foundation backfilling, and compensation foundation. Its application in building energy saving is mainly because, compared with traditional insulation boards such as polystyrene board and polyurethane board, foamed concrete has fireproof performance. Therefore, mixing waste can reduce the production costs and boost the resource utilization of industrial pollutants, in line with the concept of green development.

Current fly ash foam concrete (FAFC) research mainly focuses on strength, thermal conductivity, and pore structure. Studying strength at an early stage can enhance the compressive strength of early foam concrete because the incorporation of fly ash will reduce the bubble size effect [8]. Chen et al. studied the influence of a series of manufacturing parameters, such as water–cement ratio and fly ash, on the performance of foamed concrete, and found that the finer the fly ash, the better the compressive strength of foamed concrete [9]. Mixing fly ash and silica fume will improve the connection between slurry and aggregate, increasing the compressive strength [10,11]. Mahzad et al. [12] found that although the incorporation of fly ash would reduce the short-term compressive strength, the compressive strength would increase with the increase in age; higher silica fume incorporation will also increase the compressive strength. Ji et al. [13] analyzed that fly ash has a morphological effect and can increase the compressive strength by improving the pore structure. Adding 40% of first-grade fly ash has the best impact on pore structure improvement. Foam concrete reinforced with PP fibers positively stops microcracks and increases energy absorption and impact properties [14,15,16]. Falliano Devid et al. [17] added PP fibers to ultralight foam concrete, which increased the flexural and compressive strength by 200% and 22%, respectively. Regarding thermal conductivity, Gencel Osman [18] concluded that FC thermal conductivity decreases with increasing fly ash admixture. Seker [19] obtained thermal conductivity of 0.060~0.073 W/(m·K) for foam concrete after blending silica fume and found that the admixture of silica fume increases the thermal conductivity of foam concrete. Priyanka E. et al. [20] analyzed the effect of three different proportions of polystyrene foam particles on the thermal conductivity of FC and found that as the proportion of polystyrene foam particles increases, the thermal conductivity decreases. All the above studies indicated that silica fume, fly ash, and PP fiber effectively improve foam concrete’s thermal conductivity.

Pore structure plays a vital role in the performance of foam concrete [15,21,22,23]; porosity, pore uniformity, pore size, and pore shape factor can determine the macrostructure of FC (frost resistance, compressive strength, thermal conductivity, impermeability, sound absorption, etc.) [24,25]. The testing methods of foam concrete are mainly divided into macroscopic pore testing and microscopic pore testing. Peng et al. [26] used digital image processing technology to obtain the pore structure characteristic parameters, such as block section porosity, pore area, and its distribution. However, macroscopic inspection not only destroys the specimen but also reflects only the local information, which cannot completely obtain the overall pore distribution. With the development of technology, ultrasonic nondestructive testing is also applied to the pore structure testing of foam concrete [27,28]. Gong et al. [29] studied the relationship between slag powder and silica fume admixture on FC pore structure and frost resistance and analyzed the relationship between pore structure and frost resistance and the mechanism of freeze–thaw damage using ultrasonic nondestructive testing, image analysis, and SEM. It was found that the frost resistance and pore structure were optimal when the slag powder was 30% and the silica fume was 6%. Pang et al. [30] tested the pore structure development law of FC under different fly ash admixtures by the mercury pressure method, computer tomography, and the vacuum water retention and absorption method, and the results showed that the appropriate amount of fly ash could improve the FC pore structure. Zhang [31] et al. found that hollow microspheres can effectively enhance the pore structure of FC and significantly reduce its thermal conductivity. Liang et al. [32] studied the freeze–thaw cycle test for autoclaved aerated concrete of different densities, showing that the larger the porosity of the specimen, the greater the mass strength loss rate, the greater the damage to the specimen in the freeze–thaw cycle test, and the poorer the frost resistance of the specimen. The pore structure of foam concrete has been studied extensively. Moreover, the relationships between pore structure, mechanical properties, and thermal conductivity have been systematically demonstrated, while the durability of low-density FC has been less studied.

This study addresses the problem of the poor strength and durability of low-density FAFC, and, to improve this shortcoming, the authors conduct research through experiments for this purpose. The main work of this paper is as follows: (1) to study the effects of hydrogen peroxide, silica fume, and PP fiber dosage on the dry density, compressive strength, thermal conductivity, and different ages (7, 28, 56, 90 d) of FAFC; (2) to characterize the pore structure parameters (porosity, average pore size, pore size distribution, pore shape factor) of FAFC by using electron microscopy and image processing software and to analyze the effects of hydrogen peroxide, silica fume, and the relationship between the amount of hydrogen peroxide, silica fume, and PP fiber dosage and the variation in the FAFC pore structure; (3) based on ultrasonic testing, the effects of different PP fiber dosages on the appearance, mass loss rate, and strength loss rate of FAFC were also analyzed, and damage factors were introduced to evaluate the degree inside the test blocks and analyze the mechanism of freeze–thaw damage.

## 2. Materials and Methods

### 2.1. Chemical Compositions

The chemical composition and physical indexes of cement are shown in Table 1. Class II fly ash was produced by the Ningxia Yinchuan Thermal Power Plant, and its chemical composition and physical indexes are shown in Table 2. Silica fume was produced by the Ningxia Zhongtong Weiye Company, and its chemical composition and physical indexes are shown in Table 3. Tianjin Comio Chemical Reagent Co. produces hydrogen peroxide. The fiber diameter is 15 μm; tensile strength is 460 MPa, and length is 12 mm; the test water is tap water.

### 2.2. Matching Ratio

The appropriate amount of silica fume and PP fiber dosage can significantly improve the compressive strength and splitting tensile strength of FAFC and improve the brittleness of FAFC. Therefore, this test was designed to carry out the physical and mechanical property and pore structure test research on FAFC with three variables: hydrogen peroxide dosing, silica fume dosing, and PP fiber dosing. The specific matching ratio is shown in Table 4.

### 2.3. Specimen Preparation

Physical foaming: the foaming agent aqueous solution is converted into foam by the mechanical stirring method, and then the foam is added to the slurry composition. The principle of physical foaming is to form a double electron layer structure in the solvent by relying on a surfactant or surface active substance, and wrap the air to form bubbles. From the molecular microstructure, surfactants are composed of two distinct parts: one is an oleophilic group (also known as hydrophobic group), and the other is a hydrophilic group (also known as oleophobic group). Based on the structural characteristics of surfactants, when the surfactants are dissolved in the solvent, the hydrophilic group is attracted by water molecules, while the hydrophilic group is repelled by water molecules. In order to achieve a stable state, the surfactant is only occupied on the surface of the solution, the oleophilic base enters the gas phase, the hydrophilic base is submerged deep into the water, and the concrete foaming agent is dissolved in the water. Mechanical stirring introduces air bubbles, which leads to a single bubble foam. The key to the formation of chemically foamed concrete is that the foaming rate of the foaming agent is consistent with the setting and hardening rate of slurry, and a dynamic balance is reached. Firstly, the foaming agent is added to the slurry and properly stirred to ensure that the foaming agent is evenly dispersed. Under the action of the initiator, the foaming agent continues to undergo chemical reactions to produce gases, forming numerous, uniformly distributed independent gas sources. Then, air pressure is gradually generated in some areas around the gas source. When the gas pressure is greater than the ultimate shear stress of the slurry (the sum of viscous resistance and hydrostatic pressure), the gas source begins to expand rapidly, forming an independent bubble, and the slurry begins to expand. In the process of air inflation, due to the hydration gel materials, the pulp consistency increases, so the expansion to overcome the resistance is also increasing; at the same time, as the reaction material is consumed, the expansion of the potential power is smaller, so the process of inflation shifts from acceleration to a gentle, slow pace, and the process gradually becomes stagnant. Finally, the expansion is completed and the foamed concrete is obtained. The chemical foaming method differs from the physical foaming method, as substances react and produce a new gas foam. The main characteristic of chemical foaming is bubbles without a bubble wall, and poor stability. The hydration products of cement and other materials should be the base material and can be stable, and the difference in the pressure of the gas bubble diameter size depends on the amount of foam material, which is difficult to control. However, the chemical foaming method has great advantages in terms of the strength, water absorption, and other properties of low-density foamed concrete [33].

In this study, FAFC was prepared by the chemical foaming method. Firstly, cement, fly ash, silica fume, water reducing agent, coagulant, foam stabilizer, PP fiber, water, and hydrogen peroxide were weighed with an electronic scale according to the pre-calculated ratio and packed in the pre-prepared box for the preparation of the FAFC test blocks. Next, the test was started by mixing cement, fly ash, silica fume, foam stabilizer (calcium stearate), coagulant (lithium carbonate), and PP fiber in a high-speed disperser for 1 min. Again, the weighed water and water reducing agent were added to the previously mixed cement slurry and continued to mix for 1 min; the mixing was stopped by mixing the cement mixture attached to the cylinder wall with water with a spatula, and then continued to mix for 1 min. Then, the pre-weighed hydrogen peroxide was added and combined for 1 min. Then, we added the pre-measured hydrogen peroxide, stirred for approximately 6~7 s until the hydrogen peroxide was evenly dispersed in the cement mortar, stopped stirring, and quickly placed the stirred FAFC into pre-greased 100 mm × 100 mm × 100 mm and 30 cm × 30 cm × 3 cm molds, and covered the top with a plastic film to avoid moisture dissipation. Finally, it was immediately placed into a standard curing box and demolded after 48 h. It was placed in the standard curing box with a standard temperature of 20 ± 2 °C and relative humidity of more than 95% for curing prior to the subsequent tests.

The FAFC test blocks shown in Figure 1a,b are FAFC test blocks with dimensions of 10 cm × 10 cm × 10 cm. Figure 1c,d shows FAFC test blocks with dimensions of 30 cm × 30 cm × 3 cm. The test block shown in Figure 1a is cut flat on the surface and demolded to become the test block shown in Figure 1b. The test block shown in Figure 1c needs to be cut after demolding to become the test block shown in Figure 1d, where Figure 1a is a standard test piece, and Figure 1c is a self-made thermal conductivity film tool.

### 2.4. Test Methods

To test the performance of the prepared specimens, dry density, compressive strength, and thermal conductivity tests, an ultrasonic test, a pore structure test, and a freeze–thaw cycle test are performed.

(1)The dry density determination method is carried out according to “Foam Concrete” JG/T266-2011 [34], and the calculation formula is
*ρ* = *M*/*V* × 10^3^(1)
where *ρ* is the dry density of the FAFC specimen, kg/m^3^; *M* is the drying mass of the FAFC specimen, g; *V* is the volume of the FAFC specimen, mm^3^.(2)Compressive strength test

The compressive strength test machine was used to test the compressive strength of specimens with the size of 100 mm × 100 mm × 100 mm, referring to (JG/T266-2011) [31]. Three samples were tested for each ratio, and, finally, the average value of three pieces was taken as the final test strength.

(3)Thermal conductivity test

The CD-DR3030 thermal conductivity tester was used. The test method of thermal conductivity determination was carried out concerning the GB/T10295-2008 [35] specification. The specimen size was 30 cm × 30 cm × 30 cm, with three specimens in each group. According to the specification, the thermal conductivity tester needs to be calibrated with professional thermal conductivity reference samples before use to ensure the accuracy of the measured FAFC thermal conductivity data. The standard models from the Building Materials Industry Technical Supervision and Research Center were used for calibration. (1) The standard model is a yellow, medium alkali glass fiber resin composite plate with 30 cm × 30 cm × (25~27) mm specification and a density range of 110~130 kg/m^3^. (2) The reference plate was first dried in a drying oven at 100 °C for 8 h. After the quality was constant, the reference plate was removed, and the average thermal conductivity was measured according to the specification GB/T10294-2008 [36], measured at the average temperature of 298 K to obtain the standard thermal conductivity value of 0.0328 W/(m·K).

(4)Ultrasonic test

This test uses the non-metallic ultrasonic testing analyzer produced by Beijing Kangkorui Company; after setting the test block size parameters, the transmitting and receiving probes are attached to the surfaces of both sides of the test block; then, we press the sampling key and store the ultrasonic testing results. Each test block tested 5 points.

(5)Pore structure test

The images of the FAFC surface pore structure were taken by an electron microscope and the images of the FAFC surface were binarized using Photoshop software. The binarized images were processed by Image-Pro-Plus image processing software, and the pore structure characteristic parameters were obtained by analysis. The effect of different hydrogen peroxide dosages, silica fume dosages, and PP fiber dosages on the usual parameters of the pore structure of FAFC was studied.

(6)Freeze–thaw cycle test

Referring to the specification JGJ/T341-2014 [37], the frost resistance of FAFC test blocks was studied by conducting 25 freeze–thaw cycles and measuring the dry density, compressive strength, and ultrasonic wave velocity of FAFC test blocks after drying at the end of every 5 cycles to study the effects of PP fiber dosage and the number of freeze–thaw cycles on the freeze–thaw resistance of FAFC, and to evaluate the quality loss rate, compressive strength loss rate, and ultrasonic wave velocity. The relationship between the pore structure and the freezing resistance was analyzed to provide a reference for improving the freezing resistance of FAFC.

## 3. Results and Discussion

### 3.1. Effect of External Admixture on Dry Density

#### 3.1.1. Analysis of Silica Fume Dosing and Hydrogen Peroxide on Dry Density

As seen in Figure 2, the relationship between the dry density of FAFC and the amount of hydrogen peroxide dosage is negatively correlated. The dry density of FAFC was 241, 225, 205, 191, 180 kg/m^3^, and the reduction rates were 6.64%, 8.89%, 6.83%, 5.76% when the hydrogen peroxide dosage was 4%, 4.5%, 5%, 5.5%, and 6%, respectively. With the increase in foaming agent content, the dry density of FAFC decreased, which was consistent with the research results of Su et al. [38]. The results showed that the amount of hydrogen peroxide dosing significantly affects the dry density of FAFC because, as the amount of hydrogen peroxide dosing increased, the bubbles per unit volume of FAFC increased, the mass of cement decreased, and the dry density decreased.

The relationship between FAFC dry density and silica fume dosage is negatively correlated in Figure 2. The dry density decreased from 218 kg/m^3^ to 202 kg/m^3^ when the silica fume dosage increased from 0 to 8%, which was reduced by 7.34%; thus, it can be seen that the silica fume dosage has little effect on the dry density. This is attributed to the fact that, although, with the increase in silica fume dosing, silica fume particles filled in the pores of the FAFC slurry, increasing the compactness of the FAFC pore wall, the volcanic ash activity of silica fume promoted the secondary hydration reaction of cement, enhancing the bubble stability of the test block. The dry density of the test block is reduced, but the amount of silica fume substituted for cement is not large, so the effect of the silica fume dosage on the FAFC dry density is not significant. The content of foaming agent can significantly affect the density, and the influence of silica fume is small. However, Liu et al. [39] found that the dry density of foamed concrete increased with the increase in silica fume content, which may be related to the raw materials used for preparing foamed concrete.

#### 3.1.2. Analysis of PP Fiber Dosage’s Effect on Dry Density

As shown in Figure 3, the relationship between FAFC dry density and PP fiber dosage is negatively correlated. The dry density of the specimen decreased from 212 kg/m^3^ to 202 kg/m^3^ with a slight change of 4.72% when the PP fiber dosage increased from 0 to 0.4%.

These findings are ascribed to the admixture of PP fiber, as the three-dimensional space mesh structure built in FAFC can improve the stability of the foam in the slurry. The dry density decreased with the increasing admixture of PP fiber. Part of the PP fiber was present in the cement slurry agglomerate. In the FAFC slurry, the essential material was cement. Therefore, the FAFC slurry is referred to as cement slurry. With the introduction of additional gas, the surrounding tiny bubbles rupture and fuse into large bubbles [40], and the dry density decreases.

### 3.2. Compressive Strength Test Results

#### 3.2.1. Analysis of the Effect of Foaming Agent Admixture on Compressive Strength

Figure 4 shows that the compressive strength of FAFC at the same age decreased significantly as the amount of hydrogen peroxide increased. On the one hand, with the rise in the hydrogen peroxide admixture, the cement mass per unit volume decreases. The cementitious substances, such as hydrated calcium silicate and hydrated calcium aluminate, generated by the hydration reaction, decrease. Thus, the density of the hydration products decreases, the thickness of the pore wall decreases, and the compressive strength of the FAFC test block decreases. On the other hand, with the increase in the amount of hydrogen peroxide, the rate of bubbles generated in the test block accelerates; the number of pores per unit volume increases; the pore wall becomes thinner; the pores easily break and fuse, the formation of joint pores increases, and the (harmful) pore increase leads to more pore structure defects and lower compressive strength [41].

The compressive strength of FAFC with different hydrogen peroxide admixtures increased with the extension of the curing age. This is because fly ash is a tiny spherical particle, and its tumbling effect can improve the fluidity and uniformity of FAFC slurry, filling in the spaces between the voids of foam concrete slurry and improving the compactness of FAFC. Fly ash has volcanic ash activity, and the amorphous silica that it contains can react with cement and water to generate low alkaline hydrated calcium silicate and other cementitious substances; with the extension of the curing age, the hydration products grow, the microstructure gradually becomes denser, and the macroscopic performance shows that the compressive strength of FAFC gradually increases. However, the growth rate over 7–28 d was relatively larger than during other periods, which is due to the cement hydration reaction producing Ca(OH)_2_, fly ash in SiO_2_, Al_2_O_3_, and other reactive oxides occurring in the secondary hydration reaction, generating C-S-H, Aft, and other hydration products. With the extension of the maintenance age, the active material decreases, the rate of the secondary hydration reaction slows down, and the strength growth rate decreases [42].

#### 3.2.2. Analysis of Silica Fume Admixture’s Effect on Compressive Strength

As seen from Figure 5, the compressive strength of FAFC after adding silica fume into FAFC shows a law of increasing first and then decreasing with the increase in silica fume admixture, indicating that an admixture with the appropriate amount of silica fume is beneficial to increase the compressive strength of FAFC. This can be attributed to the fact that silica fume contains a large amount of active SiO_2_, which can react with Ca(OH)_2_ to produce the cementitious substance hydrated calcium silicate, which improves the compactness and strength of the pore wall, but, with the increasing amount of silica fume admixture, the slurry fluidity decreases, the FAFC test block encounters difficulties and internal defects increase. Additionally, the increase in the silica fume admixture led to a relative decrease in cement content. The content of Ca(OH)_2_ generated by the cement hydration reaction decreases. The content of cementitious substances such as hydrated calcium silicate and hydrated calcium aluminate is reduced, and the compressive strength of FAFC decreases.

With the extension of the maintenance age, the compressive strength of FAFC samples with different amounts of silica fume admixture all showed an increasing trend. These findings are ascribed to the silica fume’s effect on FAFC, which mainly included the microfilming effect of silica fume and the volcanic ash effect [43]. Silica fume particles with small particle sizes and large specific surface areas can fill in the pores of FAFC and improve the density of FAFC. Meanwhile, in FAFC, the hydration reaction of cement generates a large amount of Ca(OH)_2_ and active SiO_2_ and Al_2_O_3_ in silica fume to produce cementitious substances such as hydrated calcium silicate and hydrated calcium aluminate. The sufficient alkaline environment inside the slurry accelerates the volcanic ash effect of silica fume and improves the compressive strength of FAFC.

#### 3.2.3. Analysis of PP Fiber Dosage’s Effect on Compressive Strength

As shown in Figure 6, the compressive strength of FAFC of the same age tends to increase first and then decrease with the increase in fiber admixture. Similarly, Geng Ling [44] found similar laws when studying the influence of polypropylene fiber and glass fiber content on the compressive strength of ultra-light foam concrete. The reason is that when the admixture of PP fiber rose from 0 to 0.1%, the fiber built a three-dimensional spatial mesh structure in FAFC, which had the role of bridging the skeleton, protecting the foam from rupture, and improving the foam stability; the fiber and hydration products were closely connected into a whole, reducing the development of cracks, while playing the role of cutting foam, so that the pore size distribution was uniform, improving the compressive strength of FAFC. Moreover, from 0.1 to 0.4%, the fiber admixture was too large and could not be evenly dispersed in the slurry. The local aggregation phenomenon occurred, which caused the foam to rupture and the pore structure to be damaged. The hydration was uneven, resulting in a reduction in the FAFC compressive strength.

With the age extension, the compressive strength of FAFC with different amounts of PP fiber gradually increased, because, in FAFC, the slurry is wrapped with PP fibers. The active substances in the slurry undergo a hydration reaction to produce hydrated calcium silicate and other cementing substances, which are closely connected with PP fibers. After the test block sets and hardens, the connection between PP fibers and the pore wall is tighter, which reduces the generation of cracks and further increases the FAFC’s compactness. The tensile strength of PP fiber is significant. In the compressive strength test, PP fiber resisted the pressure together with the pore wall, which enhanced the compressive capacity of FAFC and increased the compressive strength.

### 3.3. Thermal Conductivity Test Results

#### 3.3.1. Analysis of the Thermal Conductivity of the Dosage Amount of Foaming Agent

From Figure 7, the thermal conductivity of FAFC tends to decrease with the increase in hydrogen peroxide dosing. When the dosage of hydrogen peroxide was 4 to 6%, the maximum reduction in thermal conductivity was 7.79%. Since the density of FAFC decreases, the pore wall becomes thinner; the foam content per unit volume inside the foam concrete increases, and the foam condenses and hardens into the pores of the foam concrete, preventing the heat from convection from propagating inside the structure, reducing the thermal conductivity, and enhancing the thermal insulation [45].

#### 3.3.2. Analysis of Silica Fume Dosage’s Effect on Thermal Conductivity

It can be observed form Figure 8 that the thermal conductivity tends to increase and then decreases with the increase in silica fume dosing. The thermal conductivity was the largest when the silica fume dosing was 4%. This is related to the pore wall compactness and pore size; the silica fume’s filling effect increases the pore wall compactness. Meanwhile, SiO_2_ in silica fume promotes the secondary hydration reaction of cement, which increases the pore wall compactness; the more significant the pore wall compactness, the higher the heat transfer capacity and the greater the thermal conductivity. At the same time, incorporating silica fume increases the number of tiny pores, facilitating heat transfer and increasing thermal conductivity. A further increase in the silica fume admixture leads to a slower reaction speed of volcanic fume and lower pore wall compactness, which is not conducive to heat transfer and leads to lower thermal conductivity, lower slurry fluidity, a larger pore size, lower heat transfer capacity of large pores, and reduced thermal conductivity [46].

#### 3.3.3. Analysis of the Thermal Conductivity of the Dosage Amount of PP Fiber

The thermal conductivity increased first and decreased with the rise in the PP fiber admixture, as is visualized in Figure 9. This is because PP fibers are uniformly dispersed in FAFC and the uniformity of the cement slurry is improved, foam stability is improved, fusion is controlled, pore sizes are relatively small, and PP fibers are better integrated into the cement slurry, which increases the thermal conductivity and pore wall compactness.

In the study, the amount of PP fiber blending increased from 0.1 to 0.4%, resulting in uneven dispersion, which led to cracking around the PP fiber; the impregnation size increased, the compactness of the pore walls decreased, and the pore structure degraded; however, since gas has lower thermal conductivity than a solid, the insulation performance improved.

### 3.4. Porous Structure Parameter Test Results

#### 3.4.1. Variation Law of Foam Agent Admixture and Pore Structure of Foam Concrete

Figure 10 shows the pore structure before and after the cross-sectional treatment when foam dosing is 4%. Figure 11 visualizes the variation rule for the porosity of the mixture for different mix ratios with respect to the amount of foaming dosing. The FAFC porosity increased significantly with the increase in blowing agent dosing in a nearly linear relationship. The porosity ranged from 82.69 to 91.48%. This is because, with the increase in the blowing agent admixture, the amount of foam in the unit volume of FAFC increases during the process of foaming and slurry setting and hardening, the mass of cementitious material relatively decreases, and the pore wall of the pore becomes thinner, which creates an increase in FAFC porosity.

As visualized in Figure 12, the average pore size of FAFC increased with the increase in the hydrogen peroxide admixture. The average pore size ranged from 694.08 to 1506.03 μm. In response to an increase in hydrogen peroxide, more foam was generated, cementitious material surrounding the foam decreased, and the foam became less stable [47].

It can easily fuse with the surrounding foam, increasing the average pore size. Moreover, the more hydrogen peroxide that was mixed, the greater the rate of foam generation and the greater the impact force; after the slurry was solidified and hardened, the average pore size increased. If the volume of fusion foam is too large, or if the average pore size is too large, the test block will collapse to a certain extent during foaming. Therefore, the amount of hydrogen peroxide should not be too large [48].

As shown in Figure 13, the hydrogen peroxide admixture affected the pore size distribution of foam concrete. As the hydrogen peroxide admixture increased, the FAFC’s pore size distribution shifted towards larger pore sizes, and a rise in the surface hydrogen peroxide admixture would result in an increase in large pore sizes. With the increasing rise in hydrogen peroxide dosage, the balance of tiny pores with a pore size less than 900 μm in FAFC gradually decreases. With the increasing hydrogen peroxide dosage, the dominant pore size interval (the pore size range with the most significant percentage of pore size) increased continuously. When the hydrogen peroxide dose was 5%, the dominant pore size interval was 900~1200 μm; when the hydrogen peroxide dose was 5.5%, the dominant pore size interval was 1200~1500 μm; and when the hydrogen peroxide dose was 6%, the dominant pore size interval was greater than 1500 μm.

The pore shape factor of FAFC tended to decrease and then increase with the increase in the hydrogen peroxide admixture, as visualized in Figure 14. By increasing the hydrogen peroxide admixture, the amount of foam increased, and a large number of foams came into contact with each other, fused into large pores, and extruded between each other; as a result, shapes tended to deform, and the pore shape factors increased.

#### 3.4.2. The Variation Law of Silica Fume Dosing on the Pore Structure of Foam Concrete

Figure 15 shows the pore structure before and after the cross-sectional treatment when silica fume is 4%. It can be observed from Figure 16 that with the increase in the silica fume admixture, the FAFC porosity showed a trend of first decreasing and then increasing, and the FAFC porosity kept falling when the silica fume admixture rose from 0 to 4%. As a result of the silica fume having a micro-aggregate filling effect and volcanic ash activity, silica fume particles filled the pores of the FAFC slurry, and the active SiO_2_ in the silica fume reacted with the cement to create gels. The substance hydrated calcium silicate gel, etc., increases the compactness of FAFC, improves the stability of the foam, and reduces the porosity of FAFC. As the silica fume content increased from 4 to 8%, the slurry’s active SiO_2_ content increased, cement content was reduced, as was the Ca(OH)_2_ content. The volcanic ash activity of silica fume was reduced, the degree of the secondary hydration reaction was reduced, the uniformity and compactness of the slurry were reduced, and the foam was easily broken and fused, increasing the FAFC porosity. Excess silica fume will lead to the destruction and fusion of part of the foam to form joint pores, which increases the FAFC porosity [48]. It is shown that the appropriate amount of silica fume in FAFC can improve its porosity.

As shown in Figure 17, the average pore size of FAFC tended to decrease and then increase with the increase in silica fume dosing. When the silica fume dosing increased from 0 to 4%, the micro-aggregate filling effect and volcanic ash effect of silica fume improved the uniformity and compactness of the FAFC slurry, enhanced the stability of the foam, caused the pore size to be small, and the average pore size decreased. When the amount of silica fume was increased from 4 to 8%, the amount of silica fume was too high, the cement content was relatively reduced, the degree of secondary hydration reaction was reduced, the hydration products were reduced, the compactness and uniformity of the cement slurry were reduced, the liquidity of the slurry was reduced, the stability of the foam was reduced, the fusion of foam occurred, and the average pore size was increased.

The effect of silica fume dosage on the pore size distribution of FAFC is shown in Figure 18. Upon increasing the silica fume dosage, the FAFC pore size distribution was approximately normal, and it first migrated in the direction of a smaller pore size, and then migrated in the direction of a larger pore size. The proportion of tiny pores with FAFC pore sizes less than 900 μm increased first and then decreased with increasing silica fume dosage. Among them, the pore size distribution range was the same when the silica fume dose was 2% and 4%, and the difference was in the fact that, when the pore size interval was <900 μm, the percentage of FAFC pores with a 4% silica fume dose was more significant than that with a 2% silica fume dose.

Figure 19 shows that with increasing silica fume dosing, the pore shape factor of FAFC decreased first and then increased; the smaller the pore shape factor, the closer the pore size is to a sphere. Therefore, when the silica fume dosing is 4%, the pore shape factor is the lowest, at 1.250, and the pore shape is the most rounded. Increasing the silica fume dosage from 0 to 4% decreased the pore shape factor continuously; due to the micro-aggregate effect of silica fume, it could fill the FAFC, which increased the uniformity of the cement paste and the volcanic ash effect, which improved the cement paste’s stability, and the tumbling effect, which decreased the pore shape factor. As the silica fume levels increased from 4 to 8%, the pore shape factor continuously increased because, with increasing silica fume levels, the cement paste’s fluidity decreased.

The foam cannot be evenly dispersed in the paste during the foaming process. Mutual aggregation leads to the fusion of foam rupture, the number of irregular foams increases, and the pore shape factor increases. Thus, mixing the correct amount of silica fume can reduce the pore shape factor of FAFC.

#### 3.4.3. The Variation Law of PP Fiber Admixture on the Pore Structure of Foam Concrete

Figure 20 shows the pore structure before and after the cross-sectional treatment when fiber dosage is 0.1%. According to Figure 21, the FAFC porosity increased and then decreased with the dose of PP fibers, but the FAFC average pore size decreased and then increased with the dose of PP fibers, and the FAFC porosity, average pore size, and shape factor were the lowest at a 0.1% dose of PP fibers.

The FAFC’s pore size distribution approximated the customary distribution law with an increase in PP fiber dosage, as shown in Figure 22 and Figure 23. The pore size distribution of FAFC tended towards a large pore size. With the increase in the PP dosage, the dominant pore size interval of FAFC first decreased and then increased. The dominant pore size interval of FAFC was always in the range of 600~900 μm when the PP fiber dosage was 0, 0.2, and 0.4%. The percentage of tiny pores with a size less than 900 μm was the largest, and the pore size distribution was reasonable. Therefore, 0.1% of PP fibers was selected.

The variation trends in the pore shape factor for the five mixture ratios are shown in Figure 24. The FAFC pore shape factor tended to decrease first and then increase with the increase in the PP fiber dosage. The pore shape factor was the smallest at 0.1% of PP fibers, which was 1.218, and the pore shape was closest to spherical. The pore shape factor decreased at a 0.1% PP fiber dosage compared to FAFC without PP fibers.

The above phenomenon occurred because the PP fibers built a three-dimensional space mesh structure in the FAFC [49], increasing the foam stability and decreasing the porosity. When the PP fiber dosage increased from 0.1 to 0.4%, the agglomeration of PP fibers appeared, the homogeneity of the cement paste decreased, the foam stability decreased, the foam fractured and fused, joint pores increased, and the porosity, average pore size, and shape factor increased. This shows that mixing the appropriate amount of PP fibers can reduce the porosity and average pore size of FAFC, but excessive amounts of PP fiber will have adverse effects.

### 3.5. Freeze–Thaw Cycle Test Results

#### 3.5.1. Analysis of Freeze–Thaw Quality Loss Rate Law

The mass loss rate of FAFC test blocks is calculated by the formula
(2)$$M_{m}=\frac{M_{0}-M_{s}}{M_{0}}\times 100\%$$
where *M_m_* is the mass loss rate of the freeze–thaw cycle test, %; *M*_0_ is the dry mass of the test block before the freeze–thaw cycle test, g; *M_s_* is the dry mass of the test block after the freeze–thaw cycle test, g.

As seen in Figure 25, the mass loss rate of the FAFC test blocks without PP fiber dosage increased with the number of freeze–thaw cycles. Mass loss reached 36.9% after 15 freeze–thaw cycles, and all the surface layers of the test blocks fell off, resulting in profound mass loss. The mass loss rate of FAFC specimens with different amounts of PP fiber gradually increased with the number of freeze–thaw cycles.

When the dosage of PP fibers was 0.4%, the mass loss rate of the FAFC test block was the smallest, indicating that the dosage of PP fibers can effectively reduce the mass loss of the test block and improve the FAFC test block freezing resistance. As a result of the freezing process, a large amount of liquid water is absorbed into the pores of foam concrete, which freezes into solid ice when exposed to freezing pressure [50,51]. FAFC’s compressive strength is low when the tensile stress is greater than the tensile strength of the pore wall. The pore wall produces microcracks, and the dense pore wall structure becomes loose as a result; in the melting process, the solid ice melts into liquid water, the hole wall shrinks, and the water enters the test block inside through the cracks in the hole wall, increasing its water content. Therefore, the mass loss rate increases with the increase in the number of freeze–thaw cycles.

When the specimens are not mixed with PP fibers, the corners of FAFC specimens are the first to be damaged, and spalling occurs. With the increase in the number of freeze–thaw cycles, the spalling phenomenon became increasingly severe, and the quality loss rate increased. Comparing the test blocks with and without PP fibers, when the PP fiber content was 0.1%, the porosity decreased, the structure compactness of the pore wall increased, the compressive strength of the pore wall increased, the moisture content inside the test block decreased, the freezing pressure caused by freezing decreased, and the freezing resistance improved. At the same time, the uniformly dispersed PP fibers were bonded with the cement paste. The PP fibers provided tensile stress for the FAFC test blocks. For the test blocks whose pore walls were damaged by freezing and swelling, the broken pieces with large cracks in the main body of the test blocks could not be completely peeled off, and they were still connected to the main body of the test blocks through the PP fibers [52]. The improvement effect of PP fibers is more evident as the number of freeze–thaw cycles increases.

As the PP fiber dosage increased from 0.1 to 0.4%, the porosity increased, the pore wall became thinner, the hydration products decreased, and the pore wall compactness decreased as well.

The FAFC pore wall was more easily destroyed in the freeze–thaw cycle test because water entered rapidly through cracks in the pore wall. The water content of the test block increased, freezing pressure increased, and the FAFC pore wall was more easily damaged. However, with an increasing amount of PP fiber, the tensile stress provided by PP fibers gradually increases. The pore wall was damaged in the freeze–thaw cycle test, and the cement paste structure around the PP fibers became loose. The test blocks fell off after crumbling, resulting in the exposure of the PP fibers, but their tensile stress kept the central part of the test blocks connected. Hence, the mass loss rate decreased with the increasing amount of PP fiber.

With a certain amount of PP fiber, the quality loss rate increases slowly with the increase in the number of freeze–thaw cycles; when the number of freeze–thaw cycles is certain, the quality loss rate decreases slowly with the addition of PP fibers, and with 0.4% of PP fiber, the quality loss rate is the lowest after 5, 10, 15, 20, and 25 freeze–thaw cycles.

#### 3.5.2. Analysis of Freeze–Thaw Compressive Strength Loss Rate Law

The following formula calculates the loss rate of the compressive strength of FAFC test blocks:(3)$$\Delta f_{c}=\frac{f_{c0}-f_{cn}}{f_{c0}}\times 100\%$$
where Δ*f_c_* is the compressive strength loss rate of the freeze–thaw cycle test, %; *f_cn_* is the compressive strength of the test block after the freeze–thaw cycle test, MPa; and *f_c_*_0_ is the compressive strength of the test block before the freeze–thaw cycle test, MPa.

Figure 26 shows that for FAFC without PP fibers, after 15 freeze–thaw cycles, the test block was severely damaged, the compressive strength loss rate reached 74.1%, and the freeze–thaw cycles were terminated. With the increase in the number of freeze–thaw cycles, the compressive strength loss rate of FAFC gradually increased. The compressive strength loss rate gradually decreased with the rise in PP fiber dosing.

Since the water in the FAFC pore is liquid, the water molecules lean together due to the hydrogen bonding force, which reduces the volume. As liquid water freezes into solid ice, the water molecules are affected by a molecular force, which reduces the hydrogen bonding force. Water enters the pores through the cracks created in the pore wall in the freeze–thaw cycle test, and the more water that enters the pores, the larger the volume expansion when the water freezes. During freeze–thaw cycles, pore wall cracks gradually develop and expand, the pore wall compactness gradually decreases, internal joint pores form, the compressive strength gradually decreases, and the compressive strength loss rate gradually increases.

As a result of the freeze–thaw cycle test on the FAFC without PP fibers, the internal pores of the test block increased, the pore wall loosened, the internal structure deteriorated, the surface cracks expanded into larger cracks, the broken pieces spalled, the compressive strength decreased rapidly, and the compressive strength loss rate increased gradually. The FAFC was mixed with the PP fibers, and the cement paste was well bonded together. The PP fibers’ crack-blocking effect prevents and disperses FAFC cracks, increases the compactness of the test block, prevents water from entering the pores, and decreases the water content in the pores [53,54].

The freezing pressure generated during freezing is slight, which can improve the ability of FAFC to resist freeze–thaw damage. At the same time, PP fibers inside FAFC can share part of the internal stress and part of the freezing pressure generated by the specimen under the temperature change, inhibiting the generation and development of cracks indicated by the pore wall and improving the resistance of the pore wall to damage. There is a three-dimensional chaotic distribution of PP fibers in FAFC, and the bonding effect with cement paste prevents the surface from spalling and increases the ability of FAFC to withstand freeze–thaw cycles. The greater the admixture of PP fibers, the greater the test block’s resistance to spalling.

PP fiber is a good substitute for polyester fiber, but when the PP fiber dosage was 0.4%, the quality loss rate was the lowest after 5, 10, 15, 20, and 25 freeze–thaw cycles; when the amount of PP fiber was 0.4%, and when the amount of PP fiber was 0.4%, the quality loss rate was the lowest after 5, 10, 15, 20, and 25 freeze–thaw cycles.

#### 3.5.3. Relationship between Ultrasonic Wave Speed and the Number of Freeze–Thaw Cycles

Under the action of freeze–thaw cycles, the FAFC test block pore wall produces micro-cracks; the cracks will have an absorption and dissipation effect on ultrasonic energy, and the propagation speed of ultrasonic waves will be reduced with the increase in defects such as cracks inside the foam concrete test block.

As the number of freeze–thaw cycles increases, the cracks inside the test block increase, and the ultrasonic wave velocity decreases [55]. Ultrasonic wave velocity can be used to analyze the ability of FAFC to resist freeze–thaw cycles; this paper uses the damage factor D to evaluate the degree of damage inside the FAFC test block from freeze–thaw cycles. The damage factor D is calculated as
(4)$$D=1-\frac{E_{n}}{E_{0}}=1-\frac{V_{n}^{2}}{V_{0}^{2}}$$
where *E_n_* is the elastic modulus of the specimen after the nth freeze–thaw cycle, MPa; *E*_0_ is the initial elastic modulus of the specimen, MPa; *V_n_* is the ultrasonic wave velocity after the nth freeze–thaw cycle, km/s; and *V*_0_ is the initial ultrasonic wave velocity of the specimen, km/s.

The ultrasonic test results of the test block before the freeze–thaw cycle are shown in Table 5. When PP fiber doping was present, the ultrasonic wave velocity increased and then decreased due to the bonding effect between the cement slurry and PP fibers, with a more uniform pore size distribution, a reduction in porosity and average pore size, an increase in FAFC wall compactness, small energy loss, and large wave velocity. With an excess of PP fibers in the test block, the foam ruptures, the joint pores grow, the porosity and average pore size increase, the internal structure of the test block becomes poor, and the ultrasonic wave speed decreases.

Therefore, the initial ultrasonic wave velocity was maximum at a 0.1% PP fiber dosage and fell with the increase in the PP fiber dosage. Nonetheless, the ultrasonic wave velocity increased compared to FAFC without a PP fiber dosage.

The effect of PP fibers on the ultrasonic velocity of FAFC after freeze–thaw cycles is shown in Figure 27. In addition to the increase in cracks in the pore walls caused by the freeze–thaw cycles, the density decreases, and the ultrasonic energy loss increases as the wave passes through the defects. During the freeze–thaw cycle, the tensile stress of PP fibers resists the damage to the pore wall due to the freezing pressure. The improvement in the anti-freezing performance of PP fibers is more evident with the increase in PP fiber dosage [56]. This is consistent with the changing pattern of PP fibers on the mass loss rate and strength loss rate of FAFC, indicating that the use of ultrasonic waves could well reflect the effect of PP fibers on the internal pore structure of foam concrete after freeze–thaw cycles.

The effect of PP fibers on the damage factor of FAFC after freeze–thaw cycles is shown in Figure 28. The damage factor increased as the number of freeze–thaw cycles increased, and the damage factor decreased as the PP fiber dosing increased [57]. When the amount of PP fibers was 0.4%, the damage factor was the smallest. It increased with the freeze–thaw cycles, and decreased with high PP fiber dosing. After mixing with PP fibers, the PP fiber and cement paste were closely bonded, which increased the size of the pore wall that could resist the freezing pressure and reduced the generation and development of cracks in the pore wall under the freezing pressure. The damage to the pore structure in FAFC was reduced, and the damage factor was diminished. This conclusion is consistent with the effect of PP fiber dosing on the ultrasonic wave velocity of FAFC after the freeze–thaw cycle test, indicating that the damage factor can well characterize the impact of PP fibers on the frost resistance of FAFC.

## 4. Conclusions

Through this experimental research on the compressive strength, thermal conductivity, and durability of FAFC, the following conclusions are drawn.

(1)The test results of different hydrogen peroxide dosages on the compressive strength, thermal conductivity, and pore structure parameters of FAFC showed that the compressive strength and thermal conductivity of FAFC decreased with the increase in hydrogen peroxide dosage, and the peak compressive strength was 0.670 MPa when the hydrogen peroxide dosage was 4%. The thermal conductivity was 0.0580 W/(m·K) when the hydrogen peroxide dosage was 5%. The porosity and average pore size of FAFC are positively correlated with the hydrogen peroxide dosage, and the pore size distribution migrates in the direction of large pores. Therefore, it is recommended that the hydrogen peroxide dose is 5%.(2)In studying the effects of different silica fume dosages on the compressive strength, thermal conductivity, and pore structure parameters of FAFC at 5% of hydrogen peroxide, the dry density of FAFC showed a decreasing trend with the increase in silica fume dosage, and the remaining indexes all peaked. The peak compressive strength was 0.625 MPa, and the peak thermal conductivity was 0.0596 W/(m·K) when the silica fume dosing was 4%. The porosity, average pore size, and pore shape factor of FAFC showed a decreasing trend with the increased silica fume dosing. The pore size distribution migrates first to the small pore direction and then to the significant pore direction. Considering all the properties, the dose of silica fume is recommended to be 4%.(3)We also studied the effects of PP fiber dosage on the compressive strength, thermal conductivity, and pore structure parameters of FAFC with 5% hydrogen peroxide and 4% silica fume. The compressive strength of FAFC doped with PP fibers was better than when it was not doped. The peak compressive strength was 0.679 MPa, and the peak thermal conductivity was 0.0610 W/(m·K) when the PP fiber dosage was 0.1%. When the dosage of PP fibers was 0.1%, the porosity, average pore size, and pore shape factor of FAFC were the lowest, at 83.24%, 529.05 μm, and 1.218, respectively, and the pore size distribution first migrated to the small pore direction and then to the significant pore direction. Therefore, a PP fiber dosage of 0.1% is recommended.(4)With the growth of the number of freeze–thaw cycles, the damage index () of FAFC increased. Nonetheless, with the increase in the PP fiber admixture, its damage index () gradually decreased, and the mass loss rate, compressive strength loss rate, and damage factor of FAFC test blocks were the smallest when the admixture of PP fibers was 0.4%. Therefore, PP fibers can improve the frost resistance of FAFC.

## Figures and Tables

**Figure 1 materials-15-06077-f001:**
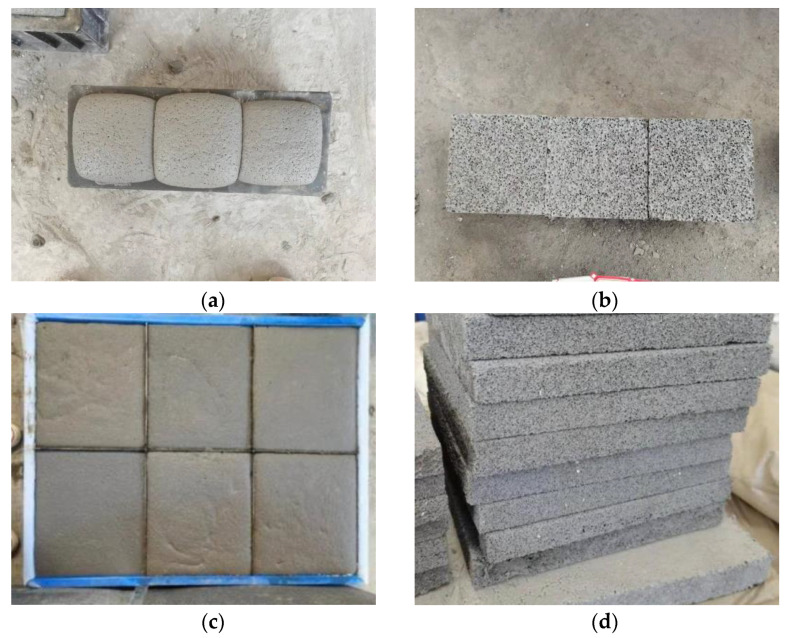
Fly ash foam concrete test block. (**a**) Test block preparation. (**b**) Test block handling. (**c**) Test block preparation. (**d**) Test block handling.

**Figure 2 materials-15-06077-f002:**
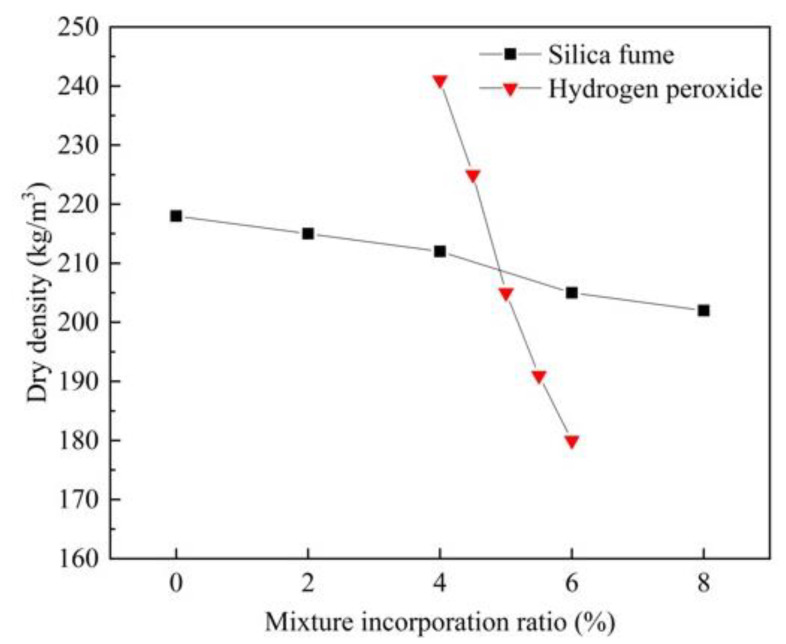
Relationship between dry density and silica fume and hydrogen peroxide dosage.

**Figure 3 materials-15-06077-f003:**
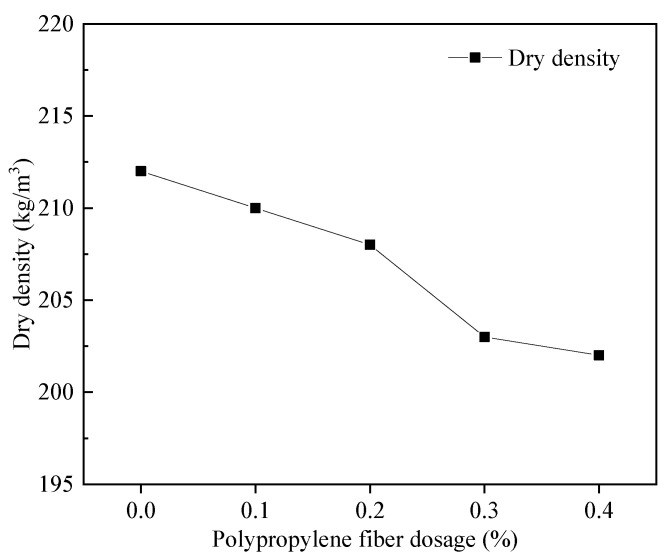
Relationship between dry density and fiber dosage.

**Figure 4 materials-15-06077-f004:**
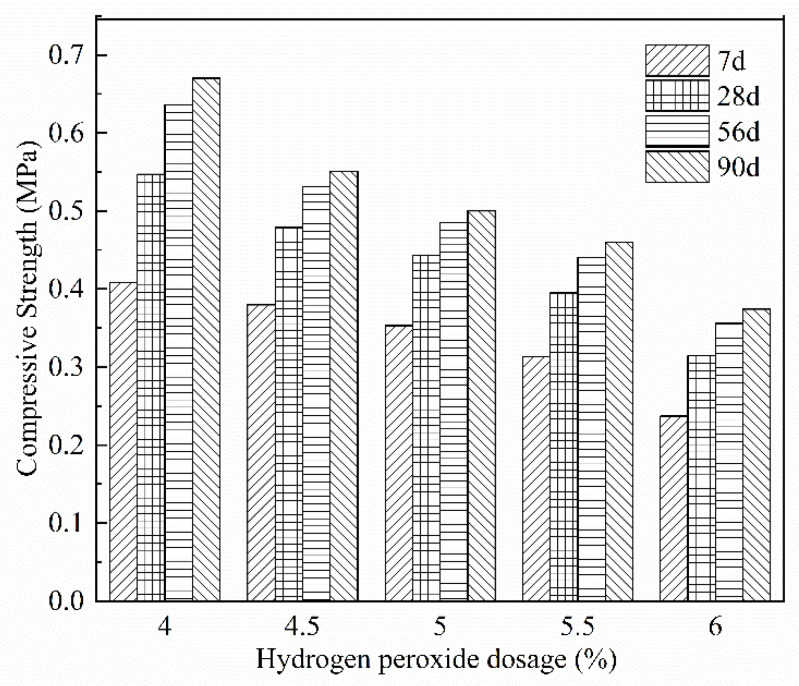
Effect of hydrogen peroxide on compressive strength at different ages.

**Figure 5 materials-15-06077-f005:**
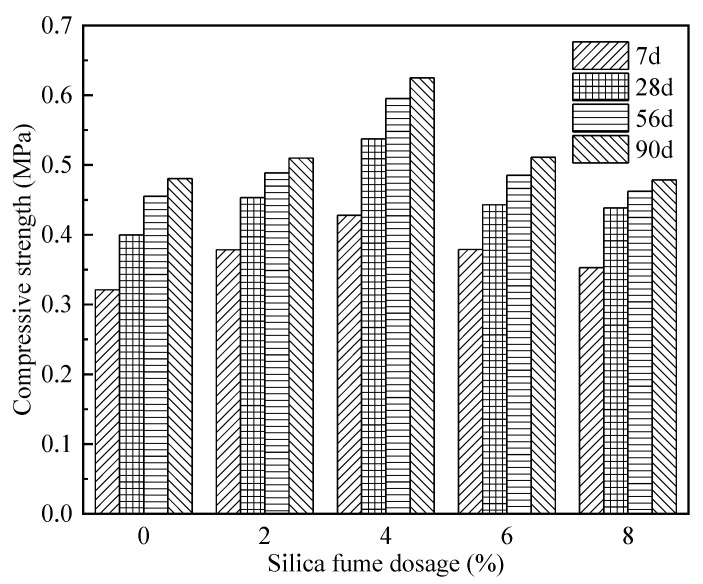
Effect of silica fume on compressive strength at different ages.

**Figure 6 materials-15-06077-f006:**
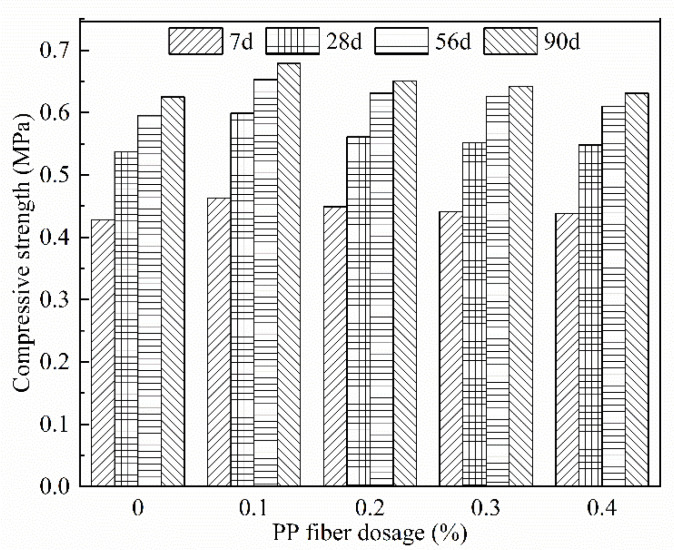
Effect of PP fiber dosage on compressive strength at different ages.

**Figure 7 materials-15-06077-f007:**
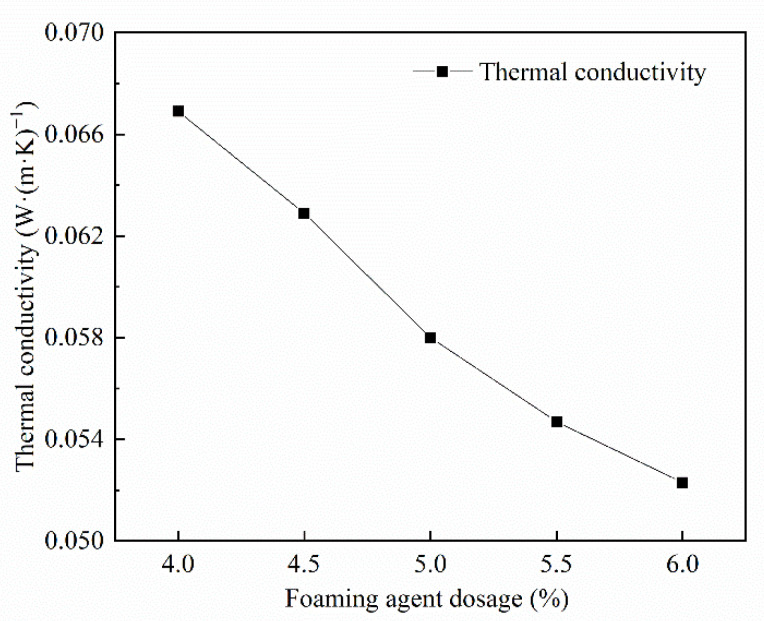
Influence of foam admixture on thermal conductivity.

**Figure 8 materials-15-06077-f008:**
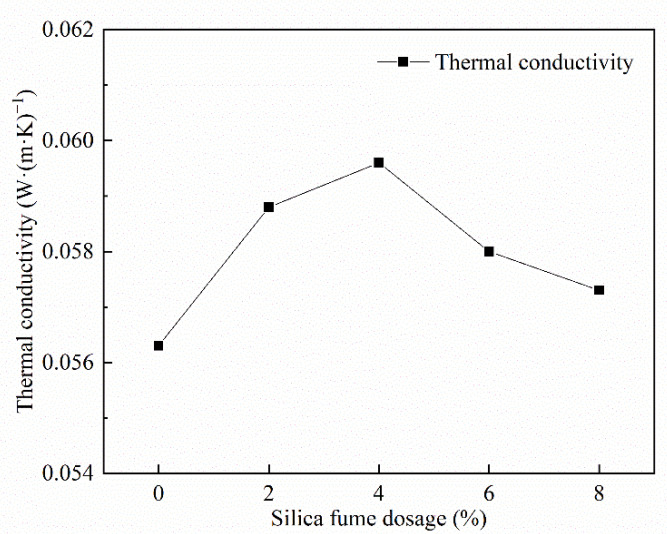
Influence of silica fume dosing on thermal conductivity.

**Figure 9 materials-15-06077-f009:**
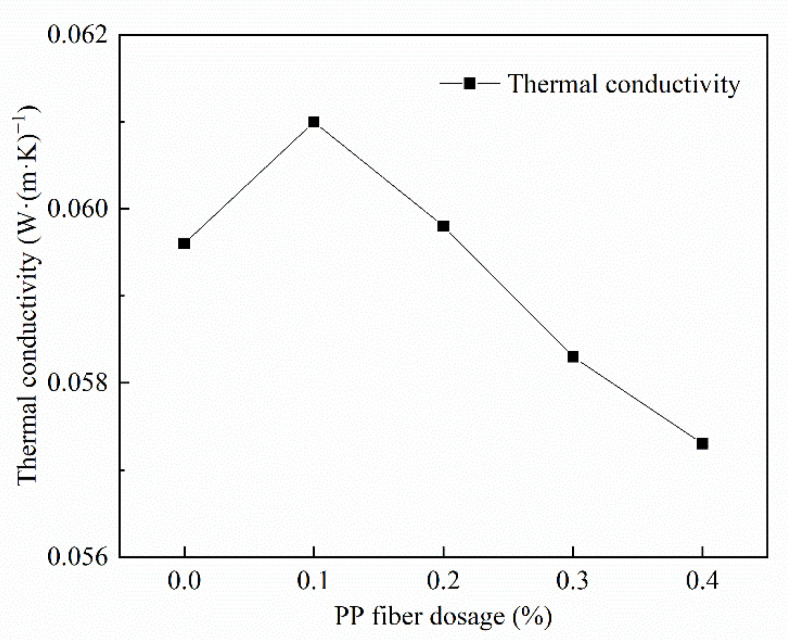
Influence of PP fiber dosage on thermal conductivity.

**Figure 10 materials-15-06077-f010:**
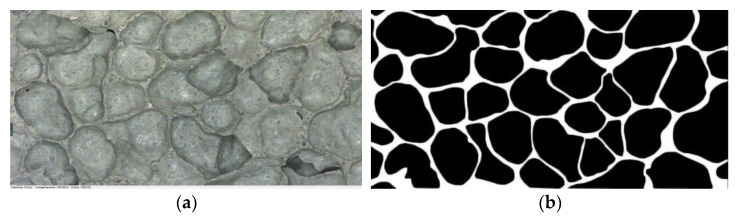
Pore cross-sectional image of 4% of foam dosing: (**a**) original image, (**b**) cross-sectional image.

**Figure 11 materials-15-06077-f011:**
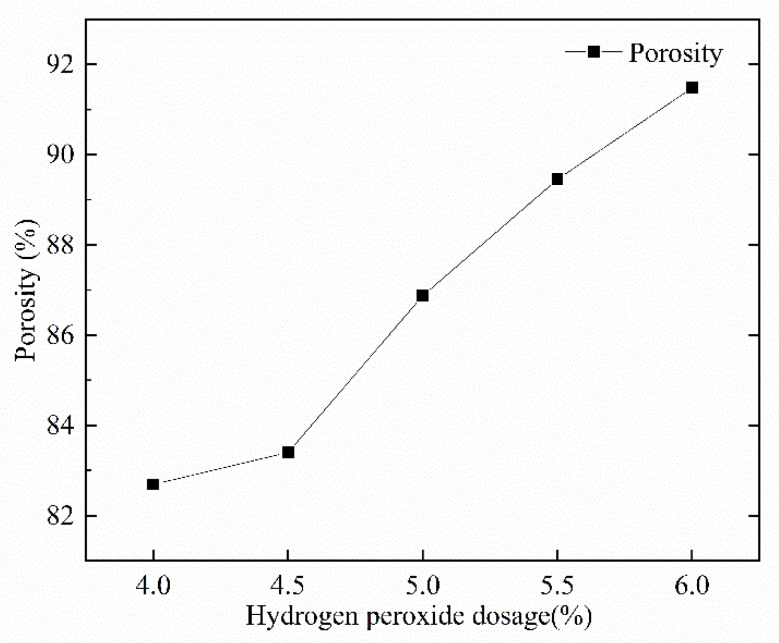
Effect of foam dosing on porosity.

**Figure 12 materials-15-06077-f012:**
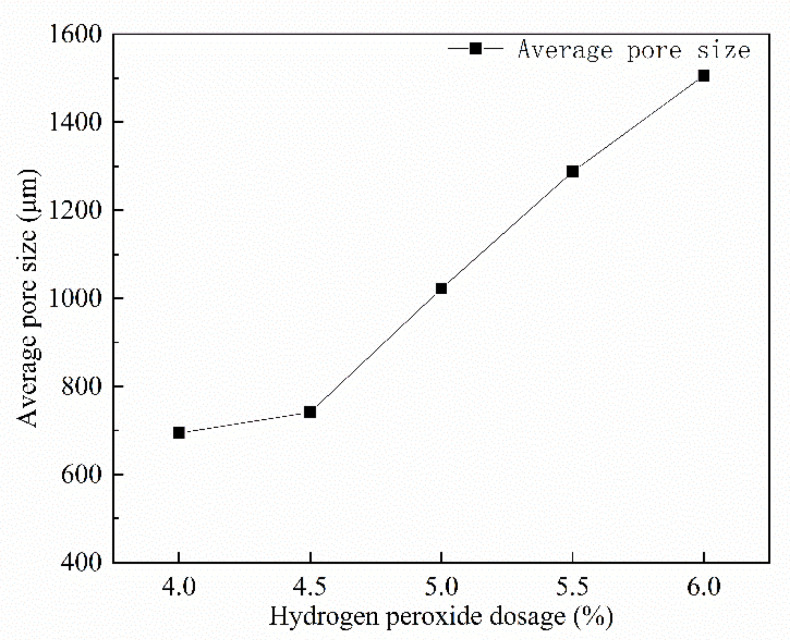
Effect of hydrogen peroxide on the average pore size.

**Figure 13 materials-15-06077-f013:**
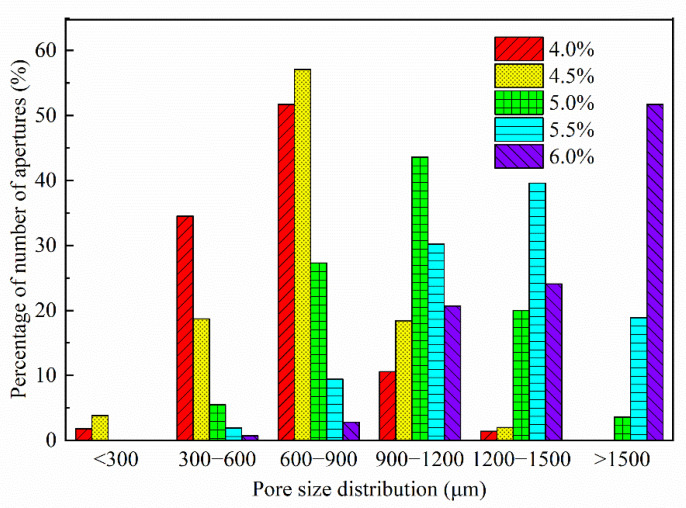
Effect of hydrogen peroxide dosing on pore size distribution.

**Figure 14 materials-15-06077-f014:**
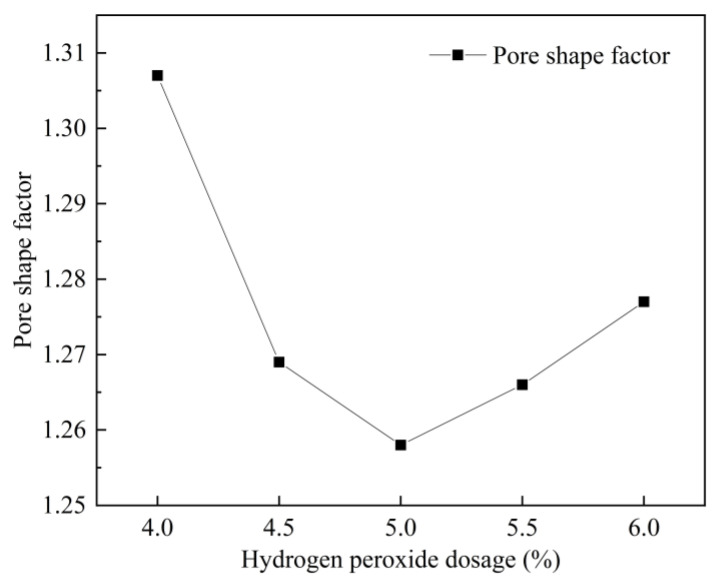
Effect of hydrogen peroxide on pore shape factor.

**Figure 15 materials-15-06077-f015:**
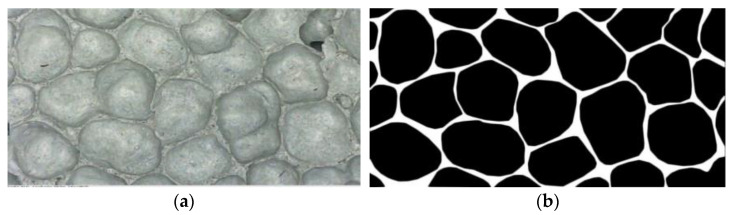
Pore cross-sectional image of silica fume dosing 4%: (**a**) original image, (**b**) cross-sectional image.

**Figure 16 materials-15-06077-f016:**
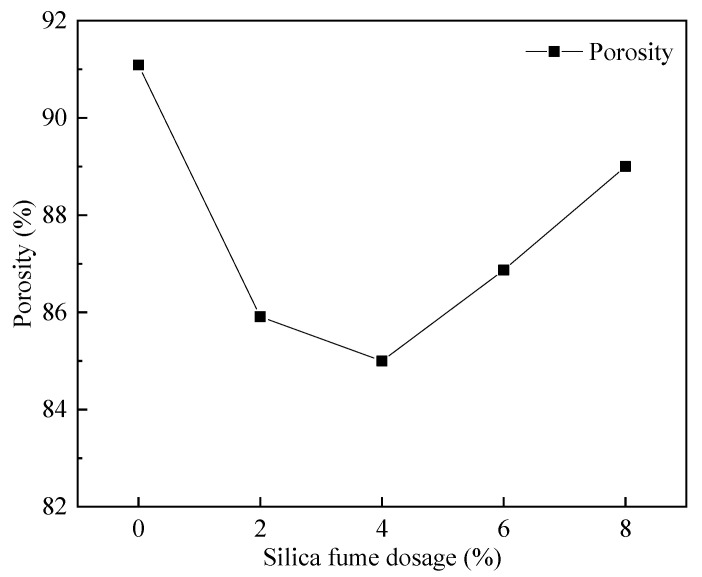
Effect of silica fume dosing on porosity.

**Figure 17 materials-15-06077-f017:**
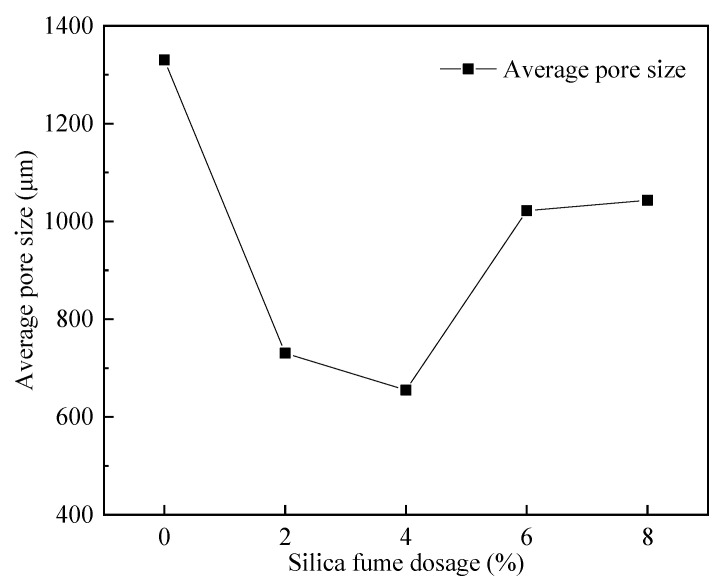
Effect of silica fume dosing on the average pore size.

**Figure 18 materials-15-06077-f018:**
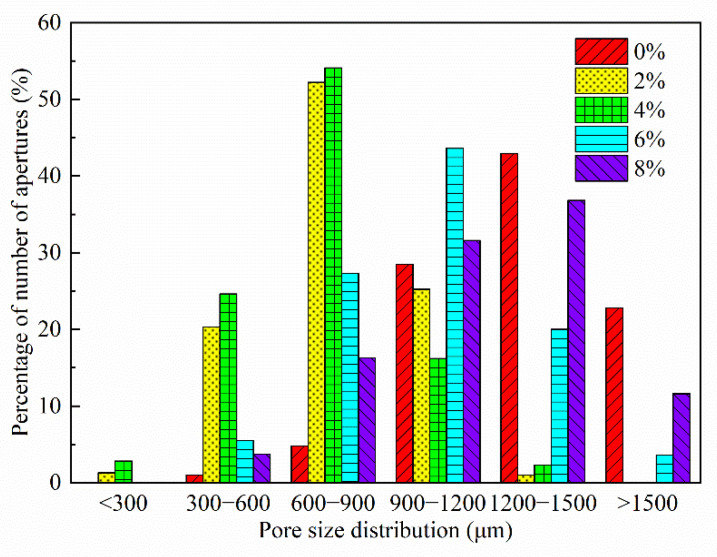
Effect of silica fume dosing on pore size distribution.

**Figure 19 materials-15-06077-f019:**
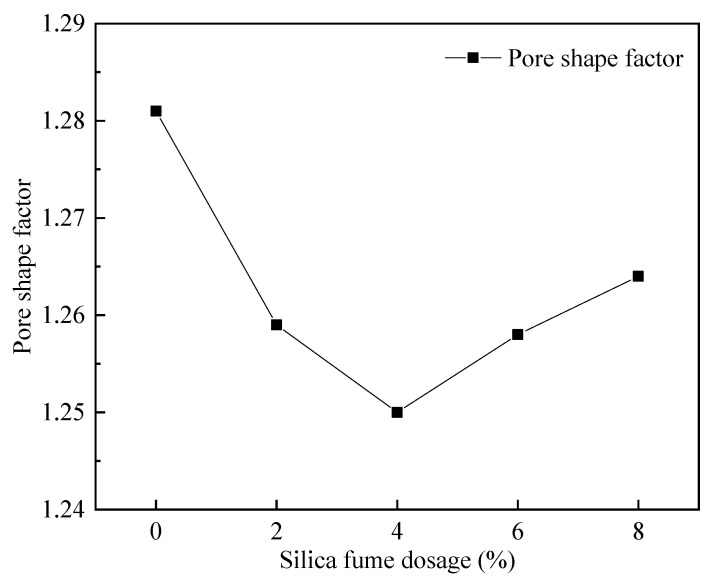
Effect of silica fume dosage on pore shape factor.

**Figure 20 materials-15-06077-f020:**
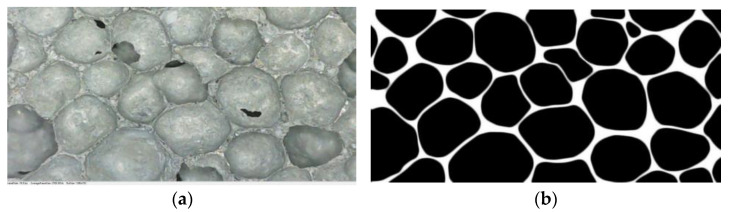
Image of pore cross-section with 0.1% fiber dosage: (**a**) original image, (**b**) cross-sectional image.

**Figure 21 materials-15-06077-f021:**
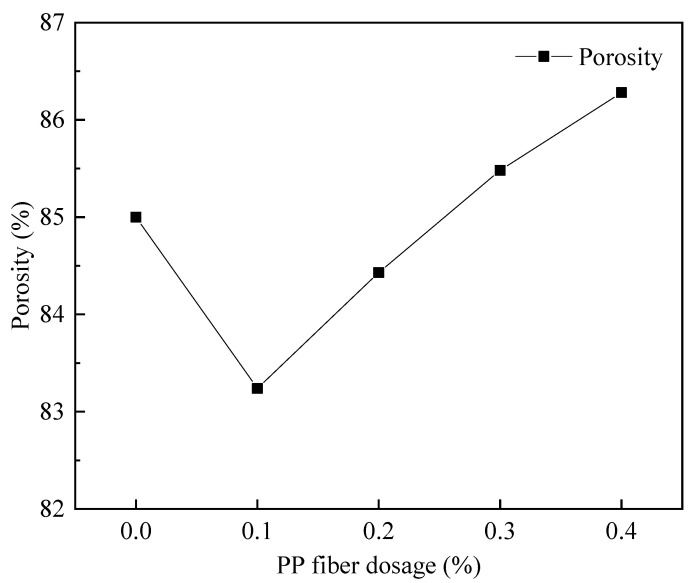
Effect of fiber doping on porosity.

**Figure 22 materials-15-06077-f022:**
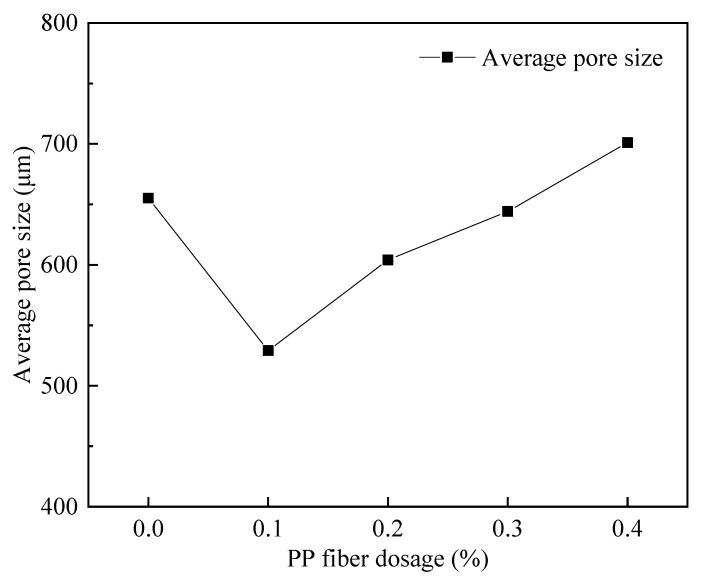
Effect of fiber doping on the average pore size.

**Figure 23 materials-15-06077-f023:**
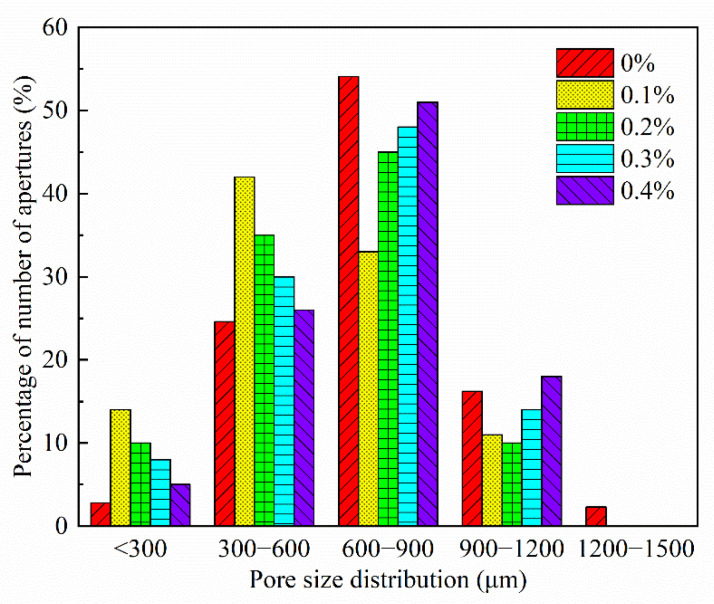
Effect of fiber doping on pore size distribution.

**Figure 24 materials-15-06077-f024:**
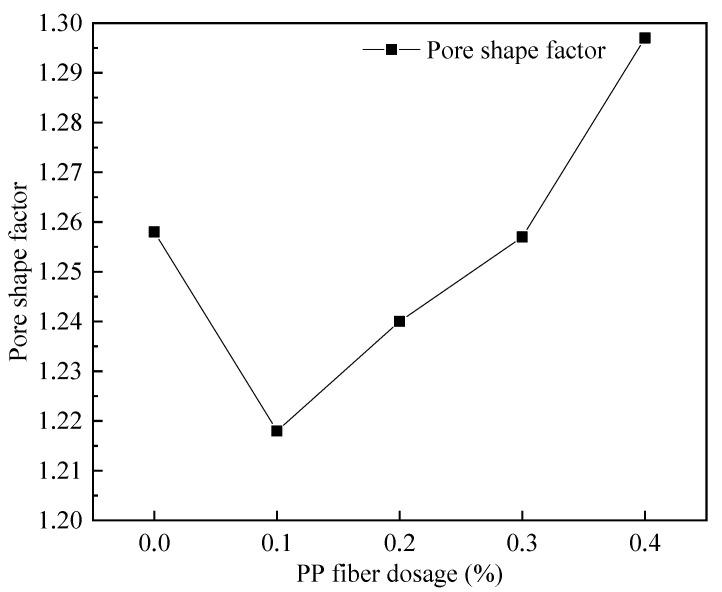
Effect of fiber doping on pore shape factor.

**Figure 25 materials-15-06077-f025:**
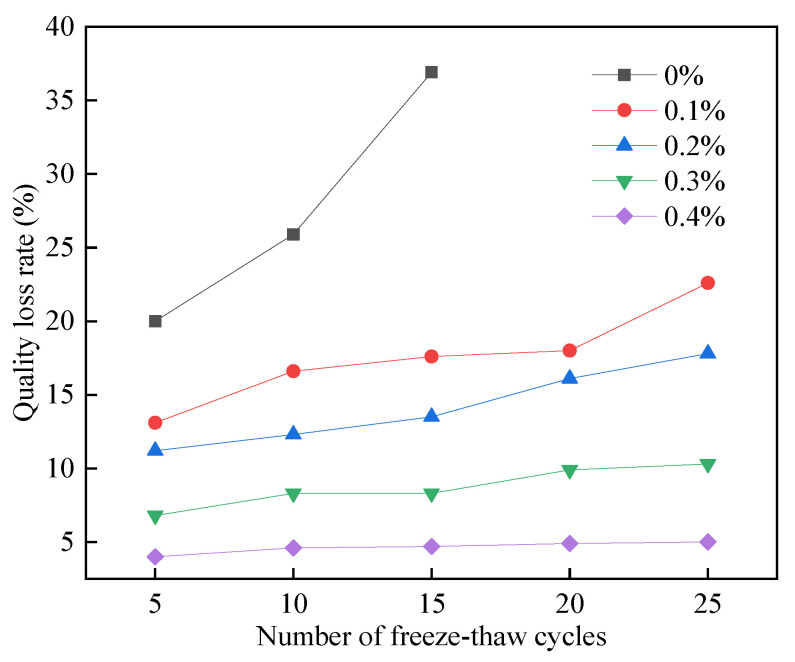
Mass loss rate of freeze–thaw cycle test.

**Figure 26 materials-15-06077-f026:**
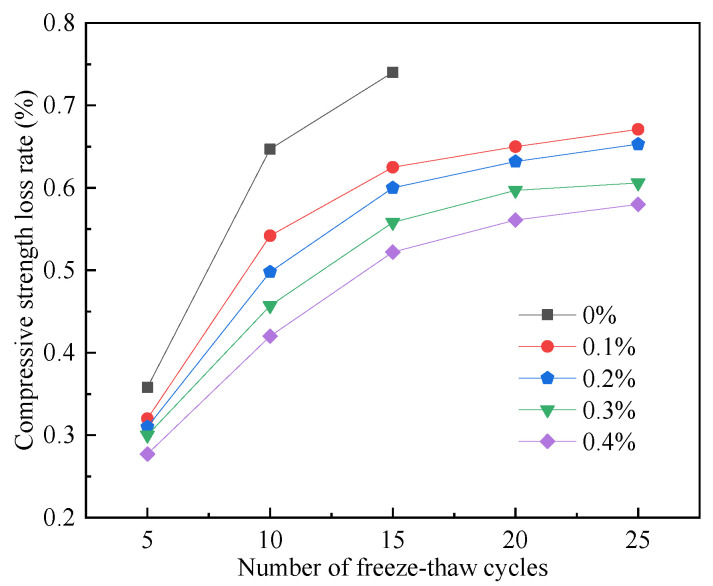
Compressive strength loss rate of freeze–thaw cycle test.

**Figure 27 materials-15-06077-f027:**
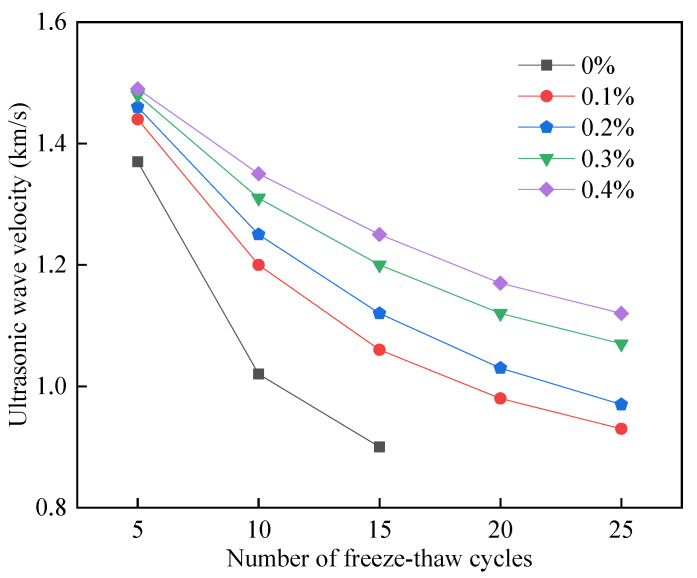
Ultrasonic wave velocity for freeze–thaw cycle test.

**Figure 28 materials-15-06077-f028:**
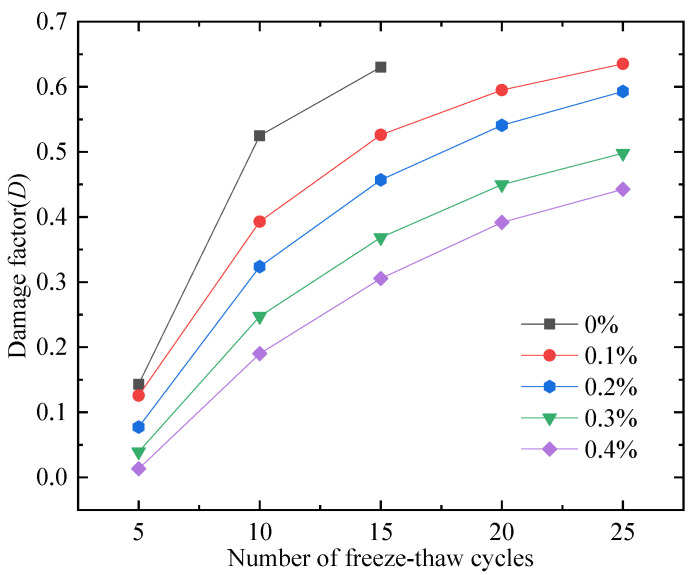
Damage factor of freeze–thaw cycle test.

**Table 1 materials-15-06077-t001:** Chemical composition and physical indicators of cement.

Chemical Composition (%)		Physical Indicators	
SiO_2_	22.36	Fineness (%)	1.6
SO_3_	2.86	Loss on ignition (%)	1.4
CaO	57.25	Coagulation time (min)	
K_2_ O	0.76	Initial condensation	200
MgO	1.44	Final condensation	245
Al_2_O_3_	7.70	Compressive strength (MPa)	
Na_2_O	0.41	3 d	25.47
Fe_2_O_3_	5.10	28 d	42.68

**Table 2 materials-15-06077-t002:** Chemical composition and physical indicators of fly ash.

Chemical Composition (%)		Physical Indicators	
SiO_2_	28.56	Fineness (%)	17.8
SO_3_	6.15		
TiO_2_	1.07	Loss on ignition (%)	2.07
CaO	12.28		
K_2_O	1.69	Water content (%)	0.33
MgO	2.10		
Al_2_O_3_	37.93	Water demand ratio (%)	97
Fe_2_O_3_	12.80		

**Table 3 materials-15-06077-t003:** Chemical composition and physical index of silica fume.

Chemical Composition (%)							Fineness (m^2^/kg)	28 d Activity Index (%)
SiO_2_	Al_2_O_3_	Fe_2_O_3_	Na_2_O	CaO	MgO	K_2_O	18,000	95
97.01	2.87	4.91	4.12	1.11	2.87	5.64		

**Table 4 materials-15-06077-t004:** Fly ash foam concrete test mix ratio.

Serial Number	Fly Ash (%)	Silica Fume (%/kg·m^−3^)	Hydrogen Peroxide (%/kg·m^−3^)	PP Fiber (%/kg·m^−3^)
1	40/83.3	6/12.5	4/8.3	0/0
2	40/83.3	6/12.5	4.5/9.3	0/0
3	40/83.3	6/12.5	5/10.4	0/0
4	40/83.3	6/12.5	5.5/11.4	0/0
5	40/83.3	6/12.5	6/12.5	0/0
6	40/83.3	0/0	5/10.4	0/0
7	40/83.3	2/4.1	5/10.4	0/0
8	40/83.3	4/8.3	5/10.4	0/0
9	40/83.3	8/16.6	5/10.4	0/0
10	40/83.3	4/8.3	5/10.4	0.1/0.2
11	40/83.3	4/8.3	5/10.4	0.2/0.4
12	40/83.3	4/8.3	5/10.4	0.3/0.6
13	40/83.3	4/8.3	5/10.4	0.4/0.8

**Table 5 materials-15-06077-t005:** Ultrasonic test results of test blocks before freeze–thaw cycles.

Test Block Proportioning	Wave Speed *v* (km/s)
0%	1.48
0.1%	1.54
0.2%	1.52
0.3%	1.51
0.4%	1.50

## Data Availability

The data supporting the findings of this study are available from the corresponding author upon reasonable request.

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
