# Peer review of "Investigating the Mechanical and Durability Characteristics of Fly Ash Foam Concrete"

_materials, 2022, doi:10.3390/ma15176077_

Round 1

Reviewer 1 Report

In the present study, the authors have been analyzed the effects of different dosing ratios of foaming agent, silica fume, and polypropylene fiber (PP fiber) on the dry density, compressive strength performance, thermal insulation performance, pore structure parameters, and durability performance of FAFC to address the problems of low strength and durability of FAFC. It is suggested to reduce the volume of the article, the number of photos and the number of pages. Extensive number of experimental test have been done in the present study which motivates to publish in the “Materials” journal.  However there are some concerns and issues that should be answered, revised and addressed in the manuscript.

Comments:

1.    Track changes should be offed in the submitted manuscript, because with this, nearly half of the page is lost!

2.    The title of the manuscript can be modified as: Investigating the mechanical and durability characteristics of fly ash foam concrete.

3.    English editing changes is required for the grammatical and syntax errors.

4.    The novelty and contribution of the present manuscript should be enhanced using following references:

-        "Utilization of circulating fluidized bed fly ash for the preparation of foam concrete." Construction and Building materials 54 (2014): 137-146.

-        "Improvement of mechanical parameters of concrete yielded from pozzolanic cement for irrigation and drainage projects." Journal of Structural and Construction Engineering 6.Special Issue 1 (2019): 43-58.

-     5.    Based on the previous comment, the title of the manuscript should be modified.

6.    Instead of using the term “foam agent” or “foaming agent”, after the first use and description of that, it is better to use the name of the chemical compound.

7.    Does the addition of “class II fly ash” alone lead to bubbles? The answer to this question should be given by reviewing technical literature and studies in which SEM results are presented.  

8.    Before the first use of abbreviated expressions, the full form of the expression must be provided. Abbreviation table is needed for the present study.

9.    The strategy of choosing the mixing design is not desirable, it is better for such studies to reduce the investigated parameters and increase the number of experiments so that in addition to ensuring the results, the results can also be used as a database: I) The number of experiments is very limited and cannot be properly analyzed and concluded; II) The selection of the range of variables is completely discrete; III) On what basis and reasoning has the value of fly ash been kept constant? IV) It is suggested that the latter cases can be mentioned as part of the limitations of the present study.

10.Any research study will be desirable when its results can be used in a practical projects. Considering the economic components, it does not seem that the result of the present study can be used in practical projects.

11.The authors should explain the process of the "chemical and physical foaming" by detail.

12.Explain the chemical formula coagulant and foam stabilizer and how they work during the preparation of samples.

13.Figures 2 and 3 should be merged and presented by different legends.

14.In Figures 5, 6, and 7, the name of the used mixing design should be presented based on the nomenclature of Table 4.

15.Figures 22 to 25 are incompletely displayed.

Author Response

Thank you for your comments. All changes are recorded in word.

Reviewer 2 Report

Comments for Authors:

- Abstract: The authors need to revise the meaning of the following sentence: "... with the increase of foaming agent dosage, the pore size distribution migrated to the direction of large pores...". The way it is written is incomprehensible. Rewrite.

- Abstract: the arguments expressed in lines 22 to 26 are confusing and poorly written. It is recommended that the authors rewrite the Abstract again.

- Abstract: at the end of the Abstract a paragraph should be added in which it is expressed in which way the results obtained could be applied in different fields of construction, sustainable materials, among others.

- Section 2. It is suggested to change the title "2. Raw materials" to "Materials and methods".

- Delete the sentence "Table 1. Chemical compositions of raw materials (%)". Repeated below. Fix.

- Table 1. I ask the authors: Who determined the chemical and physical composition of the cement used in this research? If it was determined by the cement factory, then the reference should be added. Answer.

- Tables 1 and 2. It is suggested to change the term "Burning loss" to " Loss on ignition". Fix.

- Section 3. It is proposed to the authors to change the title "Experimental results and analysis" to "Results and discussion". Fix.

- Line 217. Change "Fig. 2." to "Figure 2". Fix.

- Figure 9. The edges of the figure have to be removed. Fix.

- Figure 10. The figure is partially cut. Please delete the outer box and redraw the figure. Fix.

- Subsection 3.4. The authors entitle this subsection as: "3.4 Hole structure parameters test results". It is recommended to change the word "hole" to "porous structure". Fix.

- Figure 11. Authors put two figures, but have to add the letters "a" and "b", e.g. "Figure 11a,b". Then explain both cases at the bottom of the figure. Fix.

- Figures 12 to 15. It is not advisable to put two different figures at the same level. Fix.

- Figure 16. Authors put two figures, but have to add the letters "a" and "b", e.g. "Figure 16 a,b". Then explain both cases at the bottom of the figure. Fix.

- Figures 17 to 20. It is not advisable to put two different figures at the same level. Fix.

- Figure 21. This figure is cut. Please fix. Also, you have to add the letters "a" and "b", and explain at the bottom of this figure. Please fix.

- Figures 22 to 25. These figures are cut...!!!! Fix.

- Figure 27. The authors show several curves using the relation "Compressive strength-MPA versus freeze-thaw cycles"; different colours and symbols are used, but each sample is not identified, only the percentages are given in each case. In this way it is very difficult to interpret the results provided. Please identify the symbols in a more understandable way. Fix.

- Figure 27. Why was the sample with 0% not investigated until 25 cycles?

- The authors do not use any bibliographic citations in the discussion of the results, which makes it very difficult to demonstrate the degree of progress and novelty contributed by this work. It is recommended that a significant number of references be included in the Discussion section, as there is a great deal of previous work in this field.

Author Response

Thank you for your review, which has been modified in the text.

Round 2

Reviewer 1 Report

The reviewer is thankful to the authors for addressing the major modifications requested. At the same time the manuscript is ready to publish in the Materials journal. 

Reviewer 2 Report

The authors have restructured and improved the work in line with the observations made previously. It appears that the work now has a better presence and order, and may be of some interest to specific readers.

However, the authors are urged to further elaborate on the proposed topic and to significantly improve the writing in English.

In my opinion, it could be published in the journal Materials.